# An AI-Powered Blood Test to Detect Cancer Using NanoDSF

**DOI:** 10.3390/cancers13061294

**Published:** 2021-03-15

**Authors:** Philipp O. Tsvetkov, Rémi Eyraud, Stéphane Ayache, Anton A. Bougaev, Soazig Malesinski, Hamed Benazha, Svetlana Gorokhova, Christophe Buffat, Caroline Dehais, Marc Sanson, Franck Bielle, Dominique Figarella Branger, Olivier Chinot, Emeline Tabouret, François Devred

**Affiliations:** 1Faculté des Sciences Médicales et Paramédicales, Inst Neurophysiopathol, CNRS, INP, Aix Marseille Univ, 13005 Marseille, France; soazig.malesinski@univ-amu.fr (S.M.); DominiqueFrance.FIGARELLA@ap-hm.fr (D.F.B.); Olivier.CHINOT@ap-hm.fr (O.C.); emeline.tabouret@ap-hm.fr (E.T.); 2Faculté des Sciences Médicales et Paramédicales, Plateforme Interactome Timone, PINT, Aix Marseille Univ, 13009 Marseille, France; 3Laboratoire Hubert Curien UMR 5516, UJM-Saint-Etienne, CNRS, University Lyon, 42000 Saint Etienne, France; remi.eyraud@univ-st-etienne.fr; 4CNRS, LIS, Aix-Marseille Univ, 13009 Marseille, France; stephane.ayache@univ-amu.fr (S.A.); hamed.benazha@etu.univ-amu.fr (H.B.); 5Oracle Labs, San Diego, CA 92121, USA; anton.bougaev@oracle.com; 6MMG, INSERM, Aix-Marseille Univ, 13009 Marseille, France; svetlana.gorokhova@univ-amu.fr; 7Service de génétique Médicale, Hôpital de La Timone, APHM, 13005 Marseille, France; 8Biochemistry and Endocrinology, Hôpital de la Conception, APHM, 13005 Marseille, France; Christophe.BUFFAT@ap-hm.fr; 9MEPHI, IRD, APHM, Aix-Marseille Univ, 13274 Marseille, France; 10CNRS, UMR S 1127, Institut du Cerveau et de la Moelle épinière, ICM, Sorbonne Université, Inserm, F-75006 Paris, France; caroline.dehais@aphp.fr (C.D.); marc.sanson@aphp.fr (M.S.); franck.bielle@aphp.fr (F.B.); 11Service de Neurologie 2-Mazarin, AP-HP, Hôpitaux Universitaires La Pitié Salpêtrière—Charles Foix, F-75013 Paris, France; 12Département de Neuropathologie, AP-HP, Hôpitaux Universitaires La Pitié Salpêtrière—Charles Foix, F-75013 Paris, France; 13Service d’Anatomie Pathologique et de Neuropathologie, CHU Timone, APHM, 13005 Marseille, France; 14Service de Neuro Oncologie, Hopital de La Timone, APHM, 13005 Marseille, France

**Keywords:** glioma, biomarker, nanoDSF, diagnostic, liquid biopsy

## Abstract

**Simple Summary:**

Brain cancers, such as gliomas, are very difficult to detect because of their localization and late onset of symptoms. Here, we have developed a novel cancer detection method based on plasma denaturation profiles obtained by a non-conventional use of Differential Scanning Fluorimetry. Using blood samples from glioma patients and healthy controls, we show that their denaturation profiles can be automatically distinguished with the help of machine learning algorithms with 92% accuracy. This promising approach can now be extended to other types of cancers and could become a powerful pan-cancer diagnostic and monitoring tool requiring only a simple blood test.

**Abstract:**

Glioblastoma is the most frequent and aggressive primary brain tumor. Its diagnosis is based on resection or biopsy that could be especially difficult and dangerous in the case of deep location or patient comorbidities. Monitoring disease evolution and progression also requires repeated biopsies that are often not feasible. Therefore, there is an urgent need to develop biomarkers to diagnose and follow glioblastoma evolution in a minimally invasive way. In the present study, we described a novel cancer detection method based on plasma denaturation profiles obtained by a non-conventional use of differential scanning fluorimetry. Using blood samples from 84 glioma patients and 63 healthy controls, we showed that their denaturation profiles can be automatically distinguished with the help of machine learning algorithms with 92% accuracy. Proposed high throughput workflow can be applied to any type of cancer and could become a powerful pan-cancer diagnostic and monitoring tool requiring only a simple blood test.

## 1. Introduction

Diffuse gliomas are the most frequent and aggressive primary brain tumors in adults. Currently, no curative treatment is available despite the association of surgical resection, radiotherapy and chemotherapy as first-line treatment [1]. Another major challenge in glioma patient management is obtaining timely and precise histological and molecular characterization of the tumor in order to establish diagnosis and orient treatment. However, biopsies of these tumors could be impossible due to their deep or diffuse location or due to patient comorbidities. In these cases, treatment would be chosen based on MRI neuro-imaging characteristics that are often insufficient. Thus, non-invasive biomarkers to detect gliomas are critically needed. Moreover, because repeated biopsies to document disease evolution is impossible, the optimization of patient follow-up is also necessary. Currently, the evaluation of patients under treatment is based on MRI, steroid dose and clinical examination, which are often difficult to interpret after radiotherapy, anti-angiogenic therapy or immunotherapy [2,3]. Finally, an accurate and timely detection of the disease recurrence is crucial to optimize the therapeutic options and to improve patients’ treatment and quality of life. Thus, there is an urgent need in the neuro-oncology field to design new easy-to-use methods that are less invasive than histological examination and more efficient than neuroimaging in order to help patient diagnosis and to follow disease progression.

Over the last decade, several teams have shown that differential scanning calorimetry (DSC), a biophysical method used to study thermal denaturation of proteins [4,5], could be potentially used to detect a number of diseases including diabetes, Lyme disease, and several types of cancer [6]. Indeed, applying DSC directly to biofluids, such as serum, plasma or cerebro-spinal fluid (CSF), resulted in reproducible denaturation profiles specific to the clinical state of an individual [6]. The observed differences in denaturation profiles of biofluids from healthy controls and patients could be explained by changes in thermostability of the most abundant proteins or by the change in their relative concentrations. Independent of the underlying molecular mechanisms resulting in this change, the denaturation profile itself can therefore be used as a biomarker. We have recently demonstrated that despite the blood–brain barrier, the presence of glioblastoma induced specific changes in patients’ plasma that can be detected by differential scanning calorimetry (DSC) [7,8].

Despite very promising proof-of-concept studies, no clinical detection tool based on DSC profiling of biofluids has been developed due to incompatibility of the DSC instrument with a clinical high-throughput process. We now describe a novel method for high-throughput plasma profiling by repurposing another fundamental research method: nanoDSF (differential scanning fluorimetry). NanoDSF, which was originally designed to study protein thermostability [9,10], is based on the modifications of the intrinsic fluorescence of the macromolecules upon their thermal denaturation. In this study, we applied nanoDSF to analyze the plasma of patients affected by glioma and developed a novel AI-based method to automatically distinguish the denaturation profiles of patients from that of control individuals.

## 2. Results

### 2.1. Patient Characteristics

We conducted this study on a bicentric cohort of 84 glioma patients with a median age at diagnosis of 49.3 years (range, 19.6–77.5). Twenty-two patients (26%) presented with 1p/19q codeleted *IDH* mutated oligodendroglioma, 25 patients (31%) with *IDH* mutated astrocytoma and 37 patients (43%) with *IDH* wild-type astrocytoma (including 19 *IDHwt* glioblastomas) (see Table 1 for more detailed patient characteristics). All patients benefited from plasma collection before adjuvant treatment.

### 2.2. Plasma Profiling

Plasma samples from our cohort of 84 glioma patients and from 63 healthy controls were loaded to 24 capillary chips and then scanned using nanoDSF Prometheus NT.Plex instrument (Nanotemper, München, Germany) in order to obtain the denaturation profiles in the range from 15 to 95 °C. Raw data were exported into datasets that contained all of the nanoDSF outputs: fluorescence at 330 and 350 nm (F330 and F350), the ratio of these values (F330/F350) as well as absorbance at 350 nm (A350). Their respective first derivative were added to the datasets to emphasize their dynamic. The first derivatives of F330/F350 were plotted to visualize denaturation (Figure 1). As seen from this figure, the mean denaturation profile of the glioma patients’ plasma was drastically different from that of healthy individuals. Compared to results obtained with DSC [8,11], we observed a higher variability in nanoDSF denaturation profiles both within controls and glioma patients. The use of the advanced AI approach is thus justified to distinguish healthy and glioma profiles.

### 2.3. Plasma Profile Classification Using AI

In order to differentiate between the denaturation profiles of healthy individuals and glioma patients, we set out to design an automated way to classify the obtained profiles using an artificial intelligence (AI) approach. Moreover, automation is needed for future applications of this approach to much larger scale analysis in clinics as well as to detect possible subtle differences between subgroups of samples. We tested several machine learning algorithms [12] on the data: the classical Logistic Regression (LR), the often well-performing Support Vector Machine (SVM), the Neural Networks (NN), and two different ensemble methods: Random Forest (RF) and Adaptive Boosting (AdaBoost). These algorithms were evaluated using a leave-one-out approach where each datum is used once as a test, while the others are used to train the automatic classifiers; the obtained values are thus averages of as many experiments as there are data. The three nanoDSF outputs (F330, F350 and A350), the ratio F330/F350, and their respective derivatives, were tested independently and in all possible combinations as input for these artificial intelligence algorithms (best performances being obtained using all these elements together, all results reported in this paper correspond to this case). Table 2 shows the results obtained with the five machine learning algorithms (LR, SVM, NN, RG and AdaBoost) using the settings allowing the best observed performance on our 147 samples (84 patients and 63 controls). NN and AdaBoost algorithms had the best accuracy (above 92%), while all others achieved around 90% of correct classification. LR algorithm provided the lowest number of false positives (only four healthy individuals were wrongly classified as glioma patients), while AdaBoost was better at reducing the number of false negatives (only five glioma patients classified as healthy). The two algorithms with highest accuracy (NN and AdaBoost) had closely related small numbers of the two error types (false positives: five and six, respectively, false negatives: six and five, respectively). When the algorithm allowed it, we also tested its version that focuses on minimizing the number of false negatives in order to decrease the possibility of missing the diagnosis of glioma that could have devastating consequences given the rapidly developing nature of this disease. As seen from the Table 2, the false negative focusing version of the SVM algorithm (fnf-SVM) maintained the same level of overall accuracy as the original SVM (87% of correct classification) while obtaining just one false negative (corresponding to 1.19% of glioma patients misclassified by this algorithm). Taken together, our results show that the detection of glioma based on the denaturation profile of plasma can be efficiently automated. Moreover, combining high-throughput nanoDSF and automated data treatment by machine learning makes our approach compatible with large-scale applications in clinics for cancer detection using a simple blood test.

## 3. Discussion

Detection of cancers by a minimally invasive blood test, or “liquid biopsy”, has been a long-thought goal in the field. A number of different cancer detection methods have been tested over the past ten years, which are based on biophysical methods such as differential scanning calorimetry (DSC) [11], infrared technology (ATR-FTIR) [13] as well as on the detection and isolation of cell-free nucleic acids, extracellular vesicles and circulating tumor cells [14,15,16,17,18]. Among these, many pilot studies have previously tested DSC of biofluids as a one-step and low-cost approach for diagnosis of a great number of diseases, including several types of cancers [11,19,20], raising hopes of designing a unique pan-cancer diagnostic tool. However, despite the efforts invested in developing this approach, technical restrictions and low throughput of DSC instruments made them impossible to be transferred for wide use in clinics. In our study, we describe a major technical breakthrough allowing us to overcome these obstacles while keeping equivalent accuracy and all the advantages of a blood test. Indeed, our approach that uses the nanoDSF instrument requires minimal quantity of plasma, no need for sample preparation, and allows much faster sample handling due to disposable capillaries and high-powered fully automated data analysis using machine learning algorithms. Compared to classical DSC, our method provides a significant increase in throughput and reproducibility while decreasing the possibility of technical error.

Using nanoDSF, we showed that denaturation profiles of the glioma patients’ plasma was different from that of healthy individuals. The observed difference in the plasma denaturation profiles between the glioma and the healthy samples can be explained by the variation in thermal stability of the plasma constituents. Indeed, the plasma denaturation profiles correspond to the cumulative sum of those from the most abundant plasma proteins [11]. Since the thermal denaturation profile of a protein is an intrinsic property dependent on its structure, modifications, such as mutation, post translational modifications, or ligand binding, can significantly impact this profile. Interestingly, there was no major variability in nanoDSF denaturation profiles within controls, regardless of sex or age of the individuals. This can be explained by the fact that the composition of many biofluids, such as plasma, serum, cerebrospinal fluid, is meticulously maintained by the organism, thus resulting in a reproducible denaturation profile. Such healthy plasma equilibrium is altered in glioma patients, leading to the emergence of a different glioma-specific profile. Even though further studies are needed to identify the molecular basis of plasma changes in glioma patients, observed differences in the denaturation profile can be used as a biomarker. Increasing the number of samples will also likely help to distinguish profiles obtained from different subtypes of glioma.

Compared to previously published studies using denaturation profiles to detect cancers, our study provides a significance improvement due to application of machine learning algorithms to classify the obtained profiles in an automated way. This automatization will not only allow future large-scale studies that involve plasma samples from thousands of patients but will also enable future inter-cancer classification. Indeed, the glioma detection approach described in our study can be further extended to other cancers. As shown in Figure 2, the workflow of a universal cancer detection approach would consist of two stages. During the first (“AI training”) stage, a mathematical model, that is, a function assigning a clinical status to any DSF denaturation profile, is inferred using artificial intelligence (AI) approaches. To do so, sets of blood plasma samples from well-characterized cancer cases need to be assembled and processed by a DSF instrument to get plasma denaturation profiles. The profiles from patients and healthy individuals associated with their clinical status (healthy vs. cancer) are then used to constitute an atlas that serves as the input to train the artificial intelligence. With the increasing number of samples used to constitute the atlas, the final model is expected to be more and more precise. During the second (“Blood test”) stage, a sample from a previously untested patient is used to generate a plasma denaturation profile using a DSF. The obtained profile is then analyzed by the AI model obtained during the first stage and gives an instantaneous answer about the disease status of the individual.

## 4. Methods

### 4.1. Patients

The patient cohort consisted of 51 patients at La Timone Hospital (Marseille) from June 2009 to February 2017 and 33 patients at La Pitié Salpétrière Hospital (Paris) from November 2008 to September 2016. Eligible patients were those aged 18 years or older with newly diagnosed glioma for whom plasma samples were available at the time of diagnosis, before adjuvant treatment. Clinical evaluations were performed every cycle, and imaging evaluations were performed every two cycles. Treatment responses and disease progression were reviewed using the RANO criteria [3]. Healthy control samples were collected at the time of routine blood collection of 63 healthy volunteers (31 male, 32 female) with a median age of 34 years (range, 16–79). All patients and healthy volunteers provided written informed consent in accordance with institutional, national guidelines and the Declaration of Helsinki.

### 4.2. Plasma Samples

Blood samples from this cohort and from 63 healthy controls were collected into EDTA tubes, separated by centrifugation (2000× *g*, 10 min, 20 °C, twice) within 30 min and then stored at −80 °C. No other specific purification step was added in order not to perturb the interactome or alter the chemical state of plasma proteins. Before nanoDSF analysis, samples were thawed rapidly at 37 °C, centrifuged and loaded on a 10 μL capillary.

### 4.3. Sample Analysis by NanoDSF

Plasma samples were loaded to 10 μL capillaries and scanned using nanoDSF Prometheus NT.Plex instrument (Nanotemper) at 5% of laser power and 1 °C/min heating rate to obtain denaturation profiles in the range from 15 to 95 °C. The machine can analyze 24 samples at once; we carefully mixed patients and controls to avoid any batch effect. Raw data were exported into datasets that contained all the nanoDSF outputs: fluorescence at 330 and 350 nm (F330 and F350) as well as the ratio of these values (F330/F350) and absorbance at 350 nm (A350).

### 4.4. Algorithm Trainings

The code used was written in Python. The data preparation was carried out using the pandas library (https://pandas.pydata.org, accessed on 13 February 2021) while the machine learning algorithms were run using the scikit-learn toolbox (https://scikit-learn.org, accessed on 13 February 2021). Raw data from the nanoDSF instrument (F330, F350 and A350) were interpolated using InterpolatedUnivariateSpline from the scipy.interpolate module in order to ensure the same temperature alignment for all data. The different tested implementations are: (1) LogisticRegression from the linear_model module with parameter max_iter sets to 1000; (2) SVC from the svm module with the following combination of parameters: kernel = “poly”, gamma = “auto”, C = 1, degree between 1 and 3; kernel = “rbf”, C = 1, gamma within {0.001, 0.01, 0.1, 1, 10}. The fnf-SVM results were obtained using SVC with the same parameters except for class_weight that was set to {0:1, 1:100} (instead of the default None value); (3) MLPClassifier from the neural_network module with different architecture (reported results correspond to 3 hidden layers of 750, 200, 50, respectively), max_iter was fixed to 5000 and learning_rate = “adaptive”; (4) RandomForestClassifier from the ensemble module with parameter n_estimators fixed to 500; (5) AdaBoostClassifier from the module ensemble with a DecisionTreeClassifier from the module tree as weak classifier (parameter base_estimator) with max_depth taken between 1 and 3, n_estimators set to 100. All algorithms were evaluated using the split from the LeaveOneOut method of the model_selection module.

## 5. Conclusions

In conclusion, our proof-of-concept study demonstrates the possibility to automatically distinguish glioma patients from healthy controls by a simple blood test, using a novel technology that combines differential scanning fluorimetry and machine learning algorithms. We propose that plasma profiling using denaturation signatures by nanoDSF can be used to develop a low-cost and high-throughput pan-cancer detection method.

## Figures and Tables

**Figure 1 cancers-13-01294-f001:**
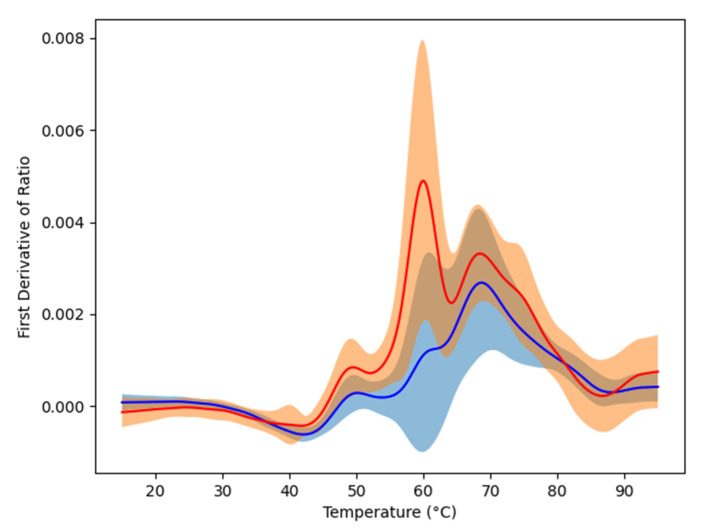
Means of the first derivatives of the ratio F330/F350. The red curve corresponds to glioma patients while the blue one is the one of the controls. The color regions graphically show the variability of the data by indicating the standard deviation.

**Figure 2 cancers-13-01294-f002:**
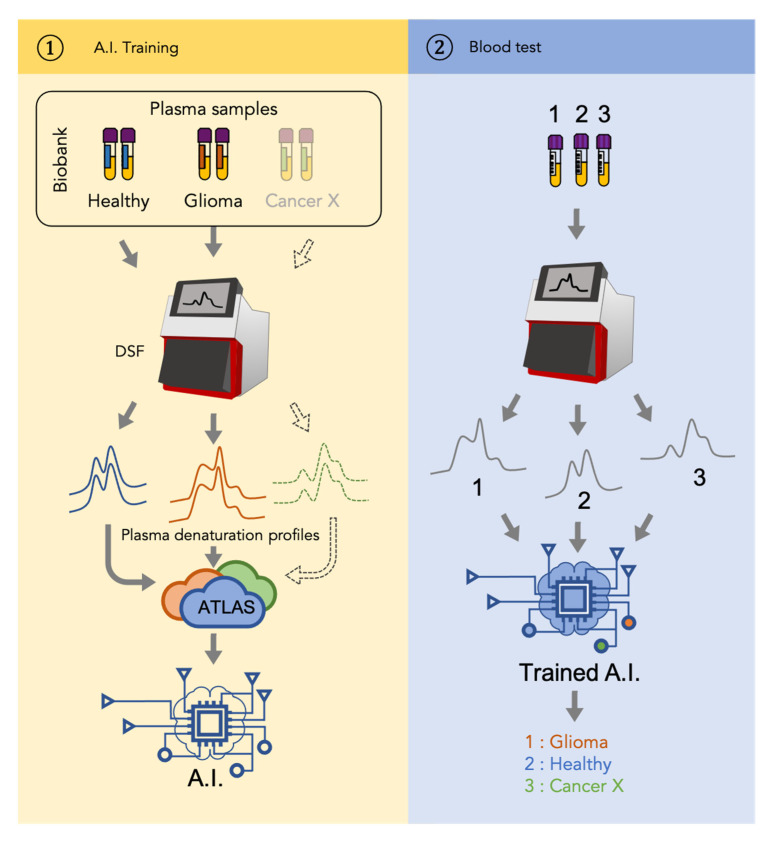
Patient-to-profile workflow/design of the study. Left panel corresponds with the training of machine learning. Denaturation profiles of plasma samples of healthy individuals and glioma patients generated by nanoDSF are added to a database (Atlas) along with their corresponding clinical status. Artificial intelligence (AI) algorithms are then trained using this Atlas to generate a model. Right panel corresponds with the use of the obtained model to identify whether the nanoDSF profile of a new sample tested corresponds with a glioma or not. Dotted line indicates that the same workflow can be applied to another cancer (Cancer X).

**Table 1 cancers-13-01294-t001:** Patient characteristics.

Factors	*N*	%
**Age** (median, min–max)	49.3 (19.6–77.5)	
**Gender** (Men/women)	45/39	54/46
**KPS** * (median, min–max)	80 (50–100)	
50	1	1
60	10	13
70	17	21
80	23	29
90	20	25
100	9	11
Steroids	51	64
**Histology**		
Oligodendroglioma		
Grade II	2	2
Grade III	20	24
Astrocytoma IDHmut *		
Grade II	3	4
Grade III	19	23
Grade IV	3	4
Astrocytoma IDHwt *		
Grade II	7	8
Grade III	11	13
Grade IV	19	22
**Surgery**		
Gross total resection	33	41
Partial resection	48	59
**Adjuvant treatment**		
Radiotherapy alone	10	12
Chemotherapy alone	12	14
Radiotherapy + chemotherapy	57	68
None	5	6

* KPS: Karnofsky Performance Scale, IDH: isocitrate dehydrogenase.

**Table 2 cancers-13-01294-t002:** Best obtained results with the different machine learning algorithms.

Algorithm	Original Algorithms					FN Minimized
	LR	SVM	NN	RF	AdaBoost	fnf-SVM
Accuracy (%)	89.80	87.07	**92.52**	89.12	**92.52**	87.07
False positives (*n*)	**4**	13	5	8	6	18
False negatives (*n*)	11	6	6	8	**5**	1
Sensitivity	0.87	0.93	0.93	0.90	0.94	**0.99**
Specificity	**0.94**	0.79	0.92	0.87	0.90	0.71
Precision	**0.95**	0.86	0.94	0.90	0.93	0.82

The best results are written in bold.

## Data Availability

The data that were used in this study to obtain the average profiles and for machine learning training are available on request from the corresponding author. The medical data are not publicly available due to ethical reasons.

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
