# Peer review of "An AI-Powered Blood Test to Detect Cancer Using NanoDSF"

_cancers, 2021, doi:10.3390/cancers13061294_

Round 1

Reviewer 1 Report

This is a very interesting manuscript with the potential of substantial utility.  I recommend acceptance of the manuscript after addressing the following comments:

  1. Citations are not in order (e.g., citations 1-3 appear, then the citations skip to 8-9, then 12, then 4-5, …). Could the authors please check throughout and amend as necessary.  Also, could the authors please check that citations appear where appropriate (e.g., citations 9, 12 in the Introduction appear to be inappropriate to the text they cite).  There are other instances where the citations appear inappropriate; please check throughout and amend as necessary.

  1. Could the authors please define KPS and IDH?

  1. Could the authors include information about the healthy controls (including the age / gender distribution)? How well matched are the controls to the glioma patients?  Could the authors also include information about the source and appropriate institutional approvals for the healthy control samples?

  1. Could the authors provide more information about how the nanoDSF parameters were tested to develop the AI algorithms? In section 2.3, the authors state that “The three NanoDSF outputs (F330, F350 and A350) were tested independently and in combination as input for these artificial algorithms”.  However, in section 2.2, it is stated that “Raw data were exported into datasets that contained all of the nanoDSF outputs: fluorescence at 330 and 350 nm (F330 and F350), the ratio of these values (F330/F350) as well as absorbance at 350 nm (A350).”  It is not clear what specific data were used as input for the AI algorithms.  What do the authors mean by “in combination”?  Was this each pair of the three NanoDSF outputs (F330, F350 and A350) as well as all three parameters together?  Did this include F330/F350, i.e., were there different combinations of four (instead of three) parameters tested?  Was the first derivative used or only the raw data?  The discussion states that “Using NanoDSF, we showed that denaturation profiles of the glioma patients’ plasma was drastically different from that of healthy individuals.”  Does this refer to Figure 1, which shows clear differences over the temperature range 45-65C based on the ratio F330/F350?  This suggests that perhaps just the F330/F350 raw data would be sufficient to discriminate the clinical groups rather than a more complex algorithm.  Did the authors compare direct use of the F330/F350 ratio with results using the different AI algorithms?

  1. Table 2 shows the best results from the different AI algorithms. Could the authors include results of all iterations of the leave-one-out analyses to show the range in diagnostic performance for each algorithm?  Sensitivity, specificity and overall accuracy could be easily plotted as box plots.  As a follow-up to the earlier point about the combination of input parameters used in the algorithms, could the authors please include the results for all combinations of parameters (perhaps this would be best to include as supplemental material)?  It would be informative to know which combinations of parameters gave the best results for each algorithm.  Were any additional analyses carried out to evaluate the discrimination of additional clinical factors (e.g., type, grade, KPS)?  If so, these additional data could also be included as supplemental material.

  1. As is common with many journals, I strongly suggest that the authors include, as supplemental data, the raw data (and any transformations made to obtain the input data for the AI analysis), all algorithms used (and their settings) and a detailed description of all analysis procedures so that readers can replicate their analyses.

Author Response

please see attachement 

Reviewer 2 Report

An extremely interesting article that opens the door to possible population screening.

My question to the authors is whether they have investigated (although I admit that perhaps the number of patients does not allow it) whether nanoDSF outputs values can show differences according to glioma grade or even according to IDH mutation status and in any case to add a comment about it in the article.

Author Response

An extremely interesting article that opens the door to possible population screening.

Thank you very much for this positive comment.

My question to the authors is whether they have investigated (although I admit that perhaps the number of patients does not allow it) whether nanoDSF outputs values can show differences according to glioma grade or even according to IDH mutation status and in any case to add a comment about it in the article. 

We have indeed considered doing so, but our cohort was neither big enough nor homogeneous between grades and types to address this question with our AI tools. We will certainly address this important question in a follow up study using a larger prospective cohort. 

As suggested, we added the sentence "Increasing the number of samples will also likely help to distinguish profiles obtained from different subtypes of glioma. " in the discussion.

Reviewer 3 Report

In the manuscript "An AI-powered blood test to detect cancer using nanoDSF" by Philipp O. Tsvetkov et al. the authors describe a cancer detection method based on plasma denaturation profiles obtained by a non-conventional use of Differential Scanning Fluorimetry. The authors have used samples from 84 glioma patients and 63 healthy controls and showed that their denaturation profiles can be automatically distinguished using machine learning algorithms. The number of samples is large, and the data obtained look quite reliable. Also, previously several teams have shown that this biophysical method of study thermal denaturation of proteins could be potentially used to detect several diseases. The authors have modified the method using a combination of differential scanning fluorimetry and machine learning algorithms. I do not see any objection to publish this manuscript in the journal “Cancers”.

Author Response

Thank you very much